# Dynamics and triggers of misinformation on vaccines

**Emanuele Brugnoli**[1,2,3]*, **Marco Delmastro**[2,4]

**1** Sony Computer Science Laboratories Rome, Joint Initiative CREF-SONY, Centro Ricerche Enrico Fermi, Rome, Italy, **2** Centro Ricerche Enrico Fermi, Rome, Italy, **3** Sapienza University of Rome, Rome, Italy, **4** Ca' Foscari University of Venice, Venice, Italy

* emanuele.brugnoli@sony.com

**Data Availability Statement:** Facebook, Instagram, Twitter, and YouTube data are made available in accordance with their respective terms of use. Content IDs used in this work are available at: https://osf.io/nsmk3/. In the same repository, the

## Abstract

The Covid-19 pandemic has sparked renewed attention to the risks of online misinformation, emphasizing its impact on individuals' quality of life through the spread of health-related myths and misconceptions. In this study, we analyze 6 years (2016–2021) of Italian vaccine debate across diverse social media platforms (Facebook, Instagram, Twitter, YouTube), encompassing all major news sources–both questionable and reliable. We first use the symbolic transfer entropy analysis of news production time-series to dynamically determine which category of sources, questionable or reliable, causally drives the agenda on vaccines. Then, leveraging deep learning models capable to accurately classify vaccine-related content based on the conveyed stance and discussed topic, respectively, we evaluate the focus on various topics by news sources promoting opposing views and compare the resulting user engagement. Our study uncovers misinformation not as a parasite of the news ecosystem that merely opposes the perspectives offered by mainstream media, but as an autonomous force capable of even overwhelming the production of vaccine-related content from the latter. While the pervasiveness of misinformation is evident in the significantly higher engagement of questionable sources compared to reliable ones (up to 11 times higher in median value), our findings underscore the need for consistent and thorough pro-vax coverage to counter this imbalance. This is especially important for sensitive topics, where the risk of misinformation spreading and potentially exacerbating negative attitudes toward vaccines is higher. While reliable sources have successfully promoted vaccine efficacy, reducing anti-vax impact, gaps in pro-vax coverage on vaccine safety led to the highest engagement with anti-vax content.

## Introduction

In today's fast-paced and interconnected digital era, social media platforms have emerged as powerful tools that play a significant role in facilitating communication and widespread dissemination of information among individuals [1]. Whether it's breaking news, scientific discoveries, cultural phenomena, or political developments, social media acts as a conduit, ensuring that information reaches a wide audience instantaneously. Aside from these clear

models fine-tuned for classifying stance and topic, respectively, are also available.

**Funding:** The authors received no specific funding for this work.

**Competing interests:** The authors have declared that no competing interests exist.

benefits, such environments also facilitate the spread of unverified or misleading information, resulting in potentially harmful consequences, ranging from public panic and confusion to the shaping of public opinion [2]. In addition, the tendency of individuals to rely on information sources that align with their pre-existing beliefs, may exacerbate societal divisions, fostering echo chambers, and reinforcing existing biases [3].

In this context, health-related topics take center stage [4], often harboring divergent perspectives [5] and enduring myths [6,7], with potential profound consequences for people's quality of life [8,9]. Among them, vaccines have always been a subject on which misinformation is active and relevant [10–15] with historical roots going back to the first vaccines (the smallpox of the cow in the late 1700s [16]). Exposure to information questioning the safety and effectiveness of vaccination, for instance, may worsen people's attitudes toward vaccines and be difficult to refute [17–19]. Vaccination hesitancy has been an important public health issue even before Covid-19 [20–22], to the point of being named one of the top ten threats to global health in 2019 by the World Health Organization [23]. However, the proliferation of anti-vaccination misinformation through social media has recently given it new urgency due to the unprecedented scale of Covid-19 pandemic and the resulting need for rapid administration of the approved vaccines [24,25].

Despite the plethora of research on the prevalence of health-related misinformation on social media, the full extent of this problem remains unclear [26]. Nonetheless, there is evidence indicating a significant correlation between people's embrace of online misinformation and their intention to get vaccinated [27].

In this study, we focus on Italy to shed light on the prevalence of vaccine-related misinformation on the main social media platforms and its potential impact on vaccine hesitancy. The choice of Italy as a case study is first motivated by the fact that since 2016 it was affected by a heated discussion on the design, approval, and enforcement of the legislative framework on mandatory pediatric vaccinations [28]. Second, Italy was the first European country to be hit by Covid-19 in the early 2020, and even the first where the dramatic developments of the disease were accompanied by a rigorous discussion around vaccination, both about its urgency and its possible negative effects [29].

Despite the disintermediated nature of social network sites, in such digital environments, opinion leaders–users whose opinions wield significant influence–continue to play a pivotal role in disseminating information and shaping the behavior of their many followers [30]. Here, we identify opinion leaders by consolidating lists from independent third-party organizations (e.g., NewsGuard, Facta, Pagella Politica) and by utilizing their binary classification of news sources into either questionable (indicating a reputation for regularly disseminating misinformation) or reliable (indicating the tendency to adhere to high journalistic standards). Followers are identified as users who interact with the vaccine-related content produced by the collected sources through their social media accounts (Facebook, Instagram, Twitter, and YouTube) over the 6-year period from 2016 to 2021.

Although scholars generally converge in defining fake news as a form of falsehood intended to primarily deceive people by mimicking the look and feel of real news [31,32], when the subject discussed has a long history of misinformation campaigns (such as vaccines), questionable sources may have achieved a certain level of autonomy and misinformation may not merely represent the denial of news from reliable sources. With this respect, some recent works have shown how the lack of reliable coverage on topics of public interest may leave room for the production and dissemination of fake content [33–35]. In other words, misinformation appears to fill some of the information gaps left uncovered by professional news providers. Hence, we first adopt the Transfer Entropy approach to dynamically determine which category of sources, questionable or reliable, causally drive the agenda of the social media discussion on vaccines.

Further, drawing on state-of-the-art literature on text classification, we develop machine learning models capable of accurately inferring the stance conveyed and topic discussed in vaccine-related content written in Italian. We then apply the models to the entire corpus, aiming to characterize the perspectives offered on vaccines by both questionable and reliable sources, and investigate their correlation with user engagement, serving as a proxy for vaccine hesitancy.

Our analyses depict misinformation not merely as the denial of news from reliable sources but rather as an autonomous force within the Italian news ecosystem. We demonstrate that misinformation has been at the core of the vaccine debate for many years, with its potential impact on vaccine hesitancy underscored by a median user engagement up to 11 times higher than reliable information. Nevertheless, the ease of spreading false claims is not solely due to the presence of questionable sources but rather stems from the inability of reliable sources to effectively guide the public debate on sensitive issues over time. Understanding the temporal dynamics of public discourse is crucial to prevent it from venturing into uncontrolled spaces where unreliable information thrives. This is evident by analyzing the relationship between user engagement and the combination of stance conveyed and topic discussed in vaccine-related content. Namely, our findings highlight the critical significance of maintaining consistent and comprehensive pro-vax coverage, particularly addressing those topics where the risk of misinformation spreading and influencing negative attitudes toward vaccines is heightened. Notably, the effectiveness of vaccination, a topic well-supported by reliable sources, stands out as having the least impact from anti-vax rhetoric in terms of user engagement. Conversely, insufficient pro-vax coverage on vaccine safety aligns with heightened engagement with misinformation content promoting an anti-vax stance.

## Results and discussion

### Parasite or commensal? understanding the role of misinformation in the vaccine news ecosystem

Throughout the analyzed period (2016–2021), the evolution of the vaccine debate in Italy has undergone a few phases that emerge clearly from the data. During the first phase, the debate was particularly vibrant in 2017 when by law (Law n.119 of July 31, anticipated by the Decree Law n.73 of June 7, hereafter Vaccination Act) the Italian Government extended from four to ten the mandatory vaccinations for 0–16 years old children (anti-polio; anti-diphtheria; anti-tetanus; anti-hepatitis B; anti-pertussis; anti-Haemophilus influenzae type b; anti-morbillus; anti-rubella; anti-parotitis; and anti-varicella), and introduced fines and admission bans for unvaccinated children at school. In this regard, it should be noted that full implementation of the Vaccination Act did not occur until September 2019 due to exemptions and extensions (See Law n.108/2018).

During the last phase, the vaccine debate has been almost completely monopolized by the pandemic outbreak: first by the Covid-19 vaccine race and rollout, later by the administration of authorized vaccines and the resulting safety concerns, especially regarding the AstraZeneca vaccine (March and June 2021) leading to vaccination hesitancy [36].

The analysis of the prevalence of misinformation on vaccines reveals that a significant portion of vaccine-related information available on the four social media analyzed originates from questionable sources. This is mainly attributed to the period preceding the onset of the Covid-19 pandemic when, on average, approximately a third of vaccine-related content constituted misinformation (Fig 1 right y-axis). This result gains significance when we consider the representativeness of the analyzed source sample in relation to the Italian information landscape (See Materials and methods).

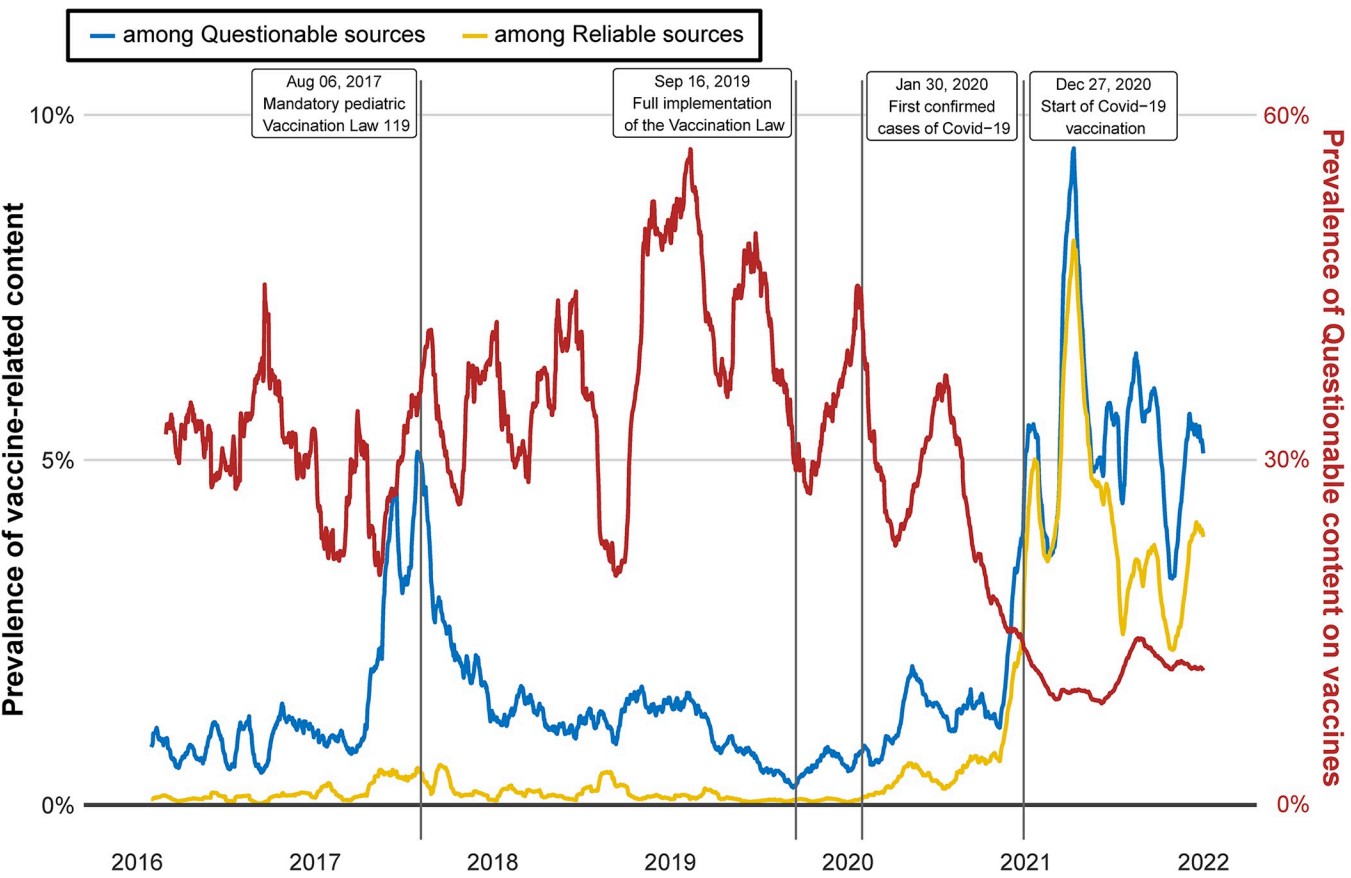

**Fig 1. Left y-axis: Daily production of vaccine-related content from questionable and reliable sources, respectively (% of total news production in the category).** Right y-axis: Daily percentage of vaccine-related content from questionable sources among all vaccine-related content produced.

The Covid-19 outbreak and the ensuing heated discussion on the vaccine race and rollout significantly heightened mainstream media attention to the vaccine subject. Consequently, the fraction of vaccine-related content from questionable sources stabilized at 10%.

Although we do not delve into potential platform effects, Facebook emerges as the primary space where unverified claims on vaccines are most widely disseminated. On average, approximately half of the vaccine-related content on the platform before the Covid-19 outbreak was indeed generated by questionable sources. The onset of the discussion on anti-Covid vaccines had a leveling effect, narrowing the differences between Facebook and the other platforms (See S2 Fig).

Fig 1 (left y-axis) also shows the prevalence of vaccine-related content among questionable (Q) and reliable (R) sources, respectively. To bring out trends more clearly, data displayed represent 30-days simple moving averages [37], i.e. the data-point at time $t$ is given by the mean over the last 30 datapoints (See S2 Table and S3 Fig for descriptive statistics of the two time-series and their corresponding first difference. See S3 Table for stationarity tests).

With this respect, the role of Covid-19 outbreak in the vaccine debate was twofold. On the one hand, regardless of the source type, it has raised media attention on vaccines to levels never reached in previous years. On the other hand, it has clearly influenced the dynamics of the cross-correlation between the two time-series. A more pronounced trend of Q than R is indeed evident before the Covid-19 outbreak, when public debate appears to have been heated almost exclusively among page communities that were skeptical about the introduction of

**Table 1. Cross-Correlation Function (CCF) between Q and R time-series with respect to the three periods analyzed.**

|  | Overall | Pre-pandemic | Pandemic |
|---|---|---|---|
| CCF | 0.840 | 0.457 | 0.905 |

mandatory vaccination. On the contrary, the two variables proceed with comparable intensity and very similar monotonicity during the pandemic (See S2 Table).

Hence, aside from performing an overall analysis of the vaccine debate in Italy throughout the time span under investigation, we identified the date of the first confirmed cases of Covid-19 in Italy (30 January 2020) as a watershed event between pre-pandemic and pandemic periods, and we also analyzed these two sub-periods separately. Note that, although the start of vaccinations dates back to 27 December 2020 when Italy received 9,750 doses of the Pfizer–BioNTech vaccine, the vaccine debate has been almost totally dominated by the Covid-19 vaccines since the early stages of the pandemic. Table 1 shows the cross-correlation function (CCF) score, i.e., the ratio of covariance to root-mean variance, between Q and R time-series with respect to any of the periods.

Consistent with what inferred graphically, these scores confirm that during the pandemic the degree of correlation between the two time-series is roughly double that of the pre-pandemic period (See S4 Fig for the lag analysis of CCF).

The different cross-correlation scores observed between the two sub-periods naturally raise questions about the drivers of the public debate on vaccines. To address these issues, we study the direct causal relationship between the two time-series by evaluating the Transfer Entropy (TE) [38] of one to the other for the overall period and both the pre-pandemic and pandemic sub-periods. TE is an information-based measure based on the Shannon's formula [39] that can appropriately detect the information flows between time-series and identify its sources. Since a straightforward implementation of TE could lead to biased estimates under conditions that may be peculiar to the observed phenomenon, we relied on the bias correction provided by the concept of Effective Transfer Entropy (ETE) [40]. The ETE estimates are reported in Table 2, together with the corresponding net information flow (NIF) from reliable to questionable, meaning that when this quantity is positive, the reliable source set informationally dominates the questionable one, whereas when it is negative, the opposite applies [41].

As far as the overall period is concerned, there is a significant bi-directional information flow between questionable and reliable source sets (1% and 5% significance level for the direction R→Q and Q→R, respectively), whereas the NIF shows a larger information transmission from the latter to the former. Hence, the results suggest that the production of vaccine news from reliable media dominates that from questionable sources.

**Table 2. Effective Transfer Entropy (ETE) estimates for both the possible information flow directions during the three analyzed periods, respectively, together with associated Standard Error (SE).** The net information flow (NIF) column represents the difference between $ETE_{R→Q}$ and $ETE_{Q→R}$.

| Period | $ETE_{R→Q}$ | SE | $ETE_{Q→R}$ | SE | NIF |
|---|---|---|---|---|---|
| Overall | 0.052*** | 0.003 | 0.012** | 0.003 | 0.040 |
| Pre-pandemic | 0.006** | 0.002 | 0.012*** | 0.002 | -0.006 |
| Pandemic | 0.047*** | 0.009 | 0.000 | 0.009 | 0.047 |

***$p < 0.001$

**$p < 0.01$

*$p < 0.05$.

**Table 3. Breakdown of the dataset.**

| CATEGORY | SOURCES | CONTENTS | | | | | | INTERACTIONS | | | | | |
|---|---|---|---|---|---|---|---|---|---|---|---|---|---|
| | | Pre-pandemic | | Pandemic | | Overall | | Pre-pandemic | | Pandemic | | Overall | |
| Questionable | 161 | 7,567 | (17.0%) | 36,980 | (83.0%) | 44,547 | (100%) | 1,801,436 | (16.5%) | 9,097,338 | (83.5%) | 10,898,774 | (100%) |
| | (23.6%) | (31.7%) | | (11.2%) | | (12.6%) | | (33.6%) | | (10.1%) | | (11.4%) | |
| Reliable | 521 | 16,293 | (5.3%) | 292,690 | (94.7%) | 308,983 | (100%) | 3,565,238 | (4.2%) | 80,766,899 | (95.8%) | 84,332,137 | (100%) |
| | (76.4%) | (68.3%) | | (88.8%) | | (87.4%) | | (66.4%) | | (89.9%) | | (88.6%) | |
| Total | 682 | 23,860 | (6.7%) | 329,670 | (93.3%) | 353,530 | (100%) | 5,366,674 | (100%) | 89,864,237 | (100%) | 95,230,911 | (100%) |
| | (100%) | (100%) | | (100%) | | (100%) | | (100%) | | (100%) | | (100%) | |

However, the breakdown of the time span into sub-periods returns misinformation not as a parasite of the news ecosystem that merely changes the object and perspective of mainstream media. Indeed, although the interactions between the two source sets are significant in both directions (1% and 5% significance level for the direction Q→R and R→Q, respectively), the information flow from R to Q undergoes a net downsizing, while it remains constant in the opposite direction, when time is limited to before the Covid-19 outbreak. Therefore, the NIF returns a slight dominance of questionable sources on reliable news media. With this respect, the very different coverage of the two source sets to the paediatric vaccination obligation, from entry into force of the Vaccination Act to its full implementation (See Fig 1 for the relative percentages and Table 3 for their numerical values), certainly played a role in determining the questionable sources as the driver of the system.

On the contrary, the Covid-19 outbreak marked a drastic increase in vaccine coverage from reliable sources to the point of decreeing their transition from dependent to independent variable in the Transfer Entropy model describing its causal relationship with the questionable counterpart. Suffice it to say that the ratio of reliable to questionable content jumped from 2 to 1 in the pre-pandemic period to 9 to 1 in the pandemic period. Moreover, although the duration of the pandemic period is roughly half that of the pre-pandemic period, questionable and reliable sources increase their overall news production on vaccines by about 500% and 1700%, respectively (See Table 3). In this new environment, the situation is practically reversed: the information flow in the direction Q→R (0.000) is not found to be statistically significant, whereas the communication from reliable sources gains its driving role in the information ecosystem and the NIF reaches its maximum (0.047), with 1% significance level for the direction R→Q.

## The engaging power of misinformation on vaccines

To determine which set of sources, questionable or reliable, has the greater impact on user engagement regarding vaccine-related content, we analyze the engagement both internally and externally. Internally, we assess how vaccine-related content compares to other topics within the same source set, while externally, we compare the engagement of vaccine-related content across questionable and reliable sources. Engagement is defined as the ratio of per-content interactions normalized by the number of followers. To compare engagement, we use a metric called the "out-engage factor" $\mathcal{P}$ (See Materials and methods), with a balance point at zero. Positive values of $\mathcal{P}$ indicate higher engagement with vaccine-related content for questionable sources (in the external comparison) and for vaccine content compared to other topics (in the internal comparison). Negative values of $\mathcal{P}$ indicate higher engagement with reliable sources (external comparison) or with other topics (internal comparison). In both cases, the content that gains more engagement in the comparison is referred to as "overperforming".

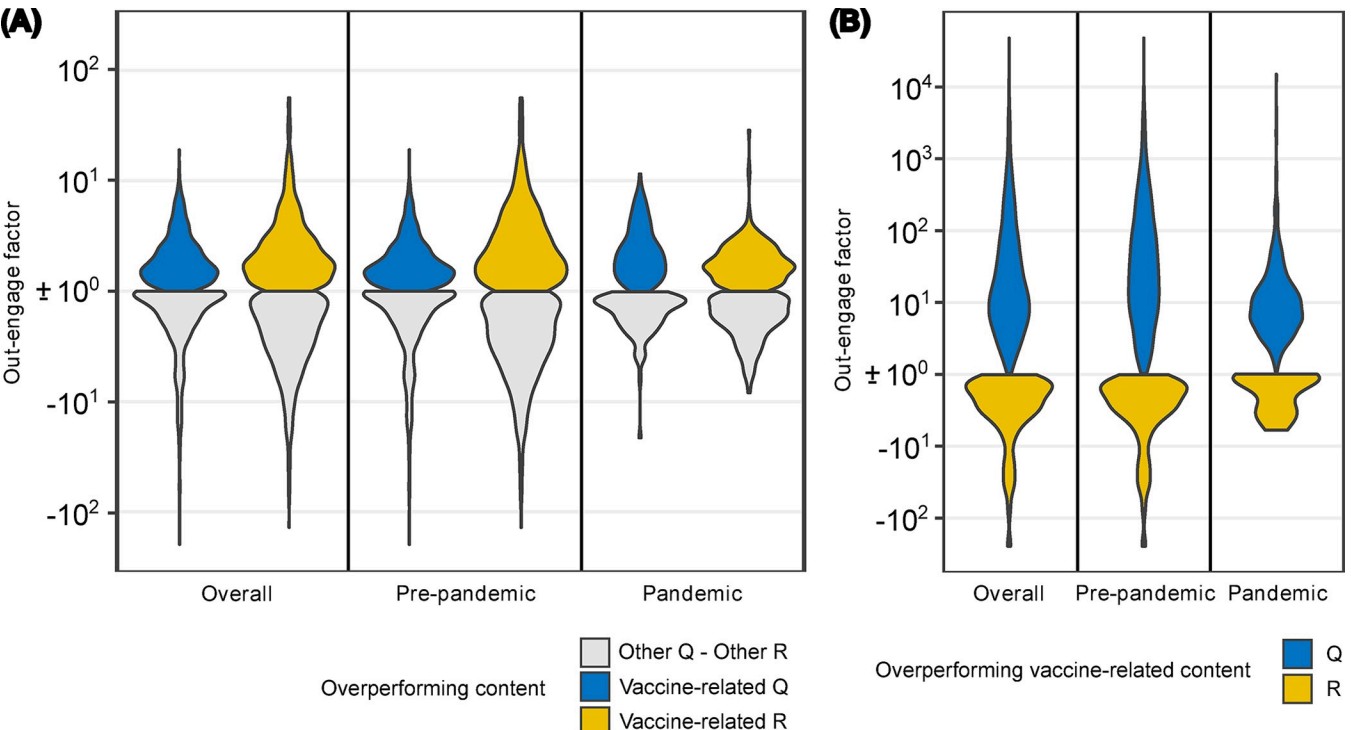

**Fig 2. Comparison of engagement with vaccine-related content between questionable (Q) and reliable (R) sources using the out-engage factor. (A) Internal comparison:** This panel shows the distribution of the out-engage factor for vaccine-related content compared to non-vaccine content within each source set (Q and R). Positive values indicate higher engagement with vaccine-related content, while negative values indicate higher engagement with other topics. The distributions are presented separately for Q and R, across different time periods (overall, pre-pandemic, and pandemic). **(B) External comparison:** This panel shows the distribution of the out-engage factor comparing vaccine-related content between Q and R. Positive values indicate higher engagement with vaccine content from questionable sources, while negative values indicate higher engagement with vaccine content from reliable sources. Distributions are broken down into the same time periods (overall, pre-pandemic, and pandemic) for a detailed comparison.

It is worth noting that while normalization by followers does not affect the out-engage factor formula in the internal comparison (it contributes twice equally but inversely), it has a huge impact on the same formula in the external comparison, either by reducing the contribution of news content from sources with a large follower base, or by amplifying the contribution of news content from sources with a small follower base [42]. This approach prevents us from the risk of confusing scale effects with the real user engagement (i.e., mainstream news media have more audience than questionable sources and therefore trigger more user interactions, all things being equal).

Denoted with $X$ the vaccine subject and with $X^C$ the totality of other subjects covered during day $d$, Fig 2A shows the distribution of the out-engage factor $\mathcal{P}(R; X, X^c; d)$ ($\mathcal{P}(Q; X, X^c; d)$) for the days $d$ when engagement is in favor of the vaccine subject compared to the rest of subjects discussed within the source set R (Q), and vice versa. Fig 2B shows instead the distribution of the out-engage factor $\mathcal{P}(Q, R; X; d)$ on vaccine subject for the days $d$ when engagement favors one source set over the other. Distributions are broken down by period analyzed.

Internally within both source sets, no substantial differences in the absolute median values of the out-engage factor are observed across the three different periods (between 0.26 and 0.64 for Q, between 0.10 and 0.21 for R). These differences are not even statistically significant for source set R (Refer to S5 Table for Mann-Whitney U test results).

Conversely, significant differences emerge when we compare the per-content engagement normalized by followers of one source set to the other (external comparison). The audience engagement distribution for questionable sources clearly dominates that for reliable sources during the overall period ($\sim 6$ times higher in median value). This is essentially due to the enormous gap observed during the pre-pandemic period, when sources set Q reached an absolute median out-engage factor $\sim 11$ times higher than R. Overall, evidence indicates that before the sudden shock of the pandemic, both the production and consumption of vaccine-related content were primarily associated with questionable sources. This could also point to the unreadiness of reliable sources to address a communication crisis such as that which has accompanied the pandemic since its early stages. Ambiguous communication about the disease origin, transmission and treatment, disjointed narratives and mixed messages about the side effects and clots caused by the AstraZeneca vaccine—just to name a few—have fostered confusion and distrust in some communities and added to the skepticism in the entire vaccination system [43]. Nevertheless, while the COVID-19 outbreak has led to an approximate halving of the absolute out-engage factor of overperforming content from the source set R (Pre-pandemic: median 2.3; Pandemic: median 1.3), questionable sources lose more than two-thirds of the engaging power during the same period (Pre-pandemic: median 26.2; Pandemic: median 8.2). See S6 Table for Mann-Whitney U test results. Hence, although vaccine news from reliable sources were never particularly outperforming in terms of engagement compared to questionable sources, the Covid-19 outbreak significantly weakened the engaging power of misinformation.

## Fighting the spread of vaccine misinformation through compelling counter-narratives

To understand the factors contributing to the observed differences in engagement between reliable and questionable source sets, we first analyze the stances conveyed in their respective vaccine-related content [44]. To this aim, we build a state-of-the-art classification model to distinguish between three different positions on vaccines: anti-vax, neutral, and pro-vax. The model is trained on a manually annotated set of contents, achieving an accuracy score of 0.88 on the evaluation set, and then applied to the entire corpus (See Materials and methods).

The left panel of Fig 3 shows a substantial time-invariance of the distribution of vaccine-related content from reliable sources among the three stance classes. As expected, the neutral perspective is dominant, exceeding 65% in both the pre-pandemic and pandemic sub-periods, followed by pro-vax opinion and a more marginal percentage of content conveying anti-vax views, which however exceeds 10% during Covid-19 outbreak. Differently, the pandemic seems to have had a significant impact on the communication strategy of questionable sources. The anti-vax perspective, which was clearly dominant throughout the pre-pandemic period, loses about 25% in favor of uplifting views during the pandemic and drops from 65% to 40% (See S5 Fig for the percentage of vaccine coverage by each news source analyzed with respect to the different stance classes).

The right panel of Fig 3 shows the distributions of the various out-engage factors $\mathcal{P}$, as defined in Eq (4), for source sets Q and R across the different stances expressed, illustrating both the internal (A) and external (B) comparisons.

For the internal comparison, where $X$ represents the vaccine subject covered through one of the three stances by source set R, and $X^C$ represents the vaccine subject covered through the other two stances by the same source set, we compare the distribution of $\mathcal{P}(\text{R}; X, X^c; d) > 0$ with that of $\mathcal{P}(\text{R}; X, X^c; d) < 0$ (See Materials and methods). Namely, we assess the

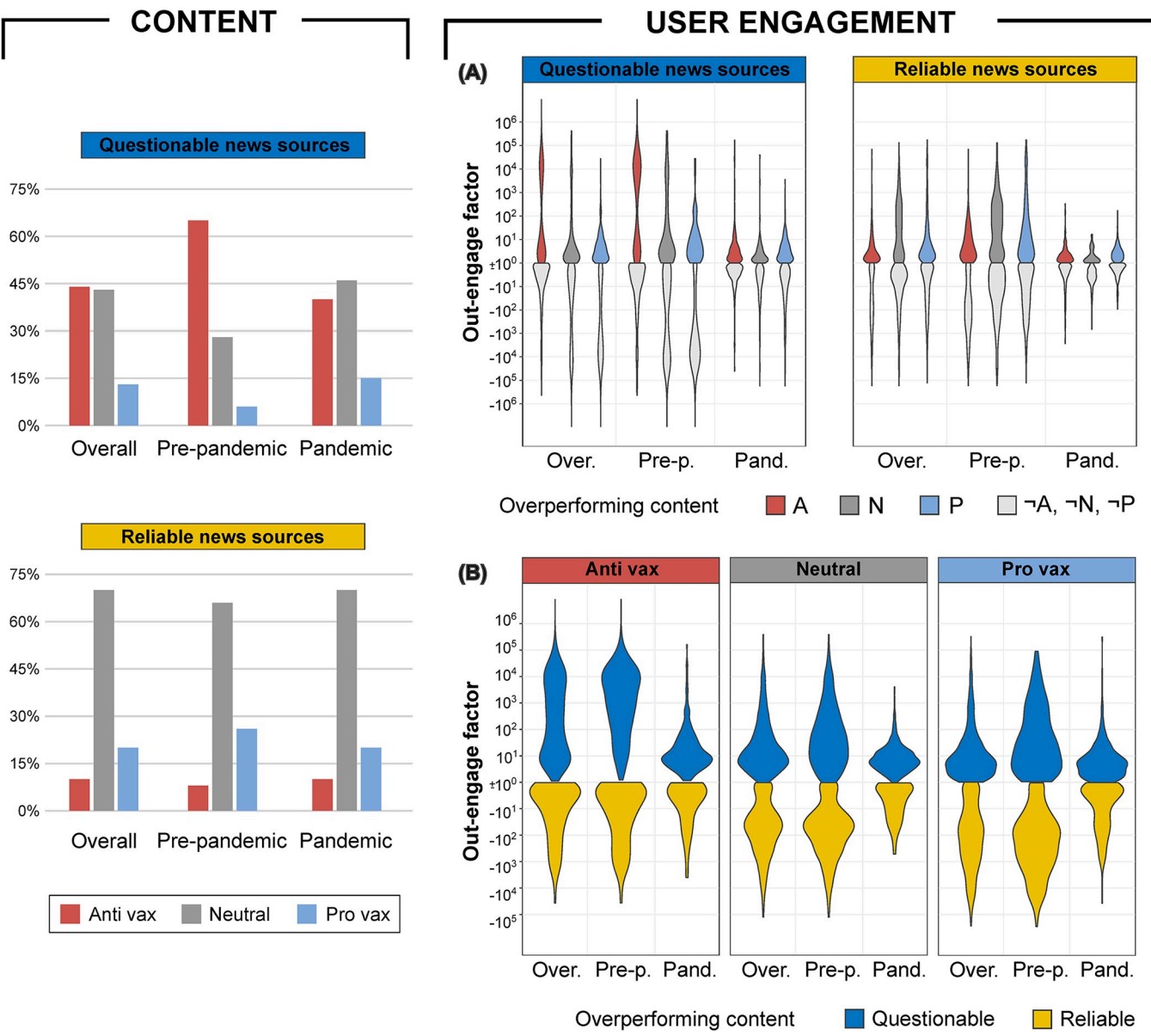

**Fig 3. Distribution of stance types and user engagement with vaccine-related content across questionable (Q) and reliable (R) sources. Left panel:** stance percentage distribution with respect to vaccine-related content from Q and R source sets, respectively. **Right panel:** distribution of out-engage factors for vaccine-related content, grouped by stance and source set, presented through both internal and external comparisons. **(A)** Internally to a source set (Q or R), the engagement with overperforming content conveying a particular stance (A = anti-vax, N = neutral, or P = pro-vax) is compared with the engagement with overperforming content conveying the other two stances (¬A, ¬N, or ¬P) within that same source set. **(B)** Externally, the same stance is evaluated across both source sets. Distributions refer distinctly to the overall period, and both the pre-pandemic and pandemic sub-periods.

engagement gained by $X$ when it overperforms $X^C$, against the engagement gained by $X^C$ when it overperforms $X$. Analogous considerations hold for source set Q.

If the most engaging vaccine-related content produced by questionable sources consistently conveys an anti-vax stance, especially before the Covid-19 outbreak, then the highly engaging content from reliable sources corresponds to neutral views before the pandemic and a pro-vax stance during the pandemic. On the contrary, uplifting views from questionable sources and anti-vax stance from reliable sources significantly underperform in terms of engagement compared to their respective dual stances. See S7 Table for Mann-Whitney U test results.

This trend is also confirmed in the external comparison, where the same stance is evaluated across both source sets. Engagement gained by content conveying anti-vax views during the overall period is notably dominated by source set Q, with a median out-engage factor approximately 40 times higher than that of source set R. Conversely, uplifting views gain greater engagement when originating from source set R (neutral median $\sim 4$ times higher and pro-vax median $\sim 15$ times higher than that of source set Q). While the differences in engagement gained from extreme positions become more pronounced when focusing on the pre-pandemic period, the sudden onset of the Covid-19 pandemic and the subsequent inundation of news about vaccines had a levelling effect, thereby aligning these metrics to comparable values (See S8 Table Mann-Whitney U test results).

In general, if anti-vax rhetoric is distinctive of questionable sources both in terms of content produced and engagement gained, such quantities are distinctive of reliable sources when expressing uplifting perspectives.

The vaccine-related contents, manually annotated with the corresponding conveyed stance, are also categorized based on the topic covered, including administration of vaccines, vaccine business, effectiveness of vaccination, legal issues, safety concerns, or other topics. This additional annotation serves as a training set for a second neural model, which is designed to distinguish between these six topics and achieve an accuracy score of 0.88 on the evaluation set (See Materials and methods).

By leveraging the results from applying both the stance and topic models to the vaccine dataset, we investigate the relationship between the discrepancy in coverage between anti-vax content from questionable sources and pro-vax content from reliable sources, and the corresponding out-engage factor for each topic. Let $\bar{C}(Q; A, \tau; T)$ and $\bar{C}(R; P, \tau; T)$ be the percentage of content on topic $\tau$ conveying anti-vax stance (A) within source set Q and pro-vax stance (P) within source set R, respectively, during period $T$. The former variable is calculated as $\Delta_{\tau}(T) = \bar{C}(Q; A, \tau; T) - \bar{C}(R; P, \tau; T)$, with $T$ spanning monthly from January 2016 to December 2021. The latter variable $\mathcal{P}_{\tau}(T)$ is derived from Eq (4) by letting $S = Q; S' = R; X = A, \tau; X' = P, \tau$. Hence, $\mathcal{P}_{\tau}(T) > 1$ if Q is overperforming compared to R and $\mathcal{P}_{\tau}(T) < -1$ vice versa.

Fig 4 shows a clear log-linear relationship between the two variables for any topic, with $R^2$ values ranging from 0.24 to 0.62 in the models $\delta(\mathcal{P}_{\tau}(T))log|\mathcal{P}_{\tau}(T)| = \alpha + \beta\Delta_{\tau}(T) + \epsilon_{\tau}(T)$, where $\delta$ denotes the sign function and $\epsilon$ the error term (See S9 Table for details on the model parameters for the various topics).

Effectiveness of vaccination and safety concerns are the topics where the corresponding fitted models exhibit both the highest slopes, $\beta = 5$ and $\beta = 3.7$, and the highest intercepts, $\alpha = 0.41$ and $\alpha = 0.64$, respectively. This indicates that the most sensitive topics are also those where the risk of misinformation spreading, and potentially exacerbating negative attitudes toward vaccines among the users involved, is higher. In this regard, reliable sources have adequately promoted the efficacy of vaccination, resulting in minimal impact from anti-vax rhetoric in terms of user engagement. Conversely, insufficient pro-vax coverage of vaccine safety has coincided with the highest engagement with misinformation conveying an anti-vax stance (See S10 Table for statistical details).

The impact of news source reliability in shaping the relationship between conveyed stance, discussed topic, and generated engagement is also explored through some econometric models. Results of the analysis are reported in S13 Table, confirming the previously discussed outcomes.

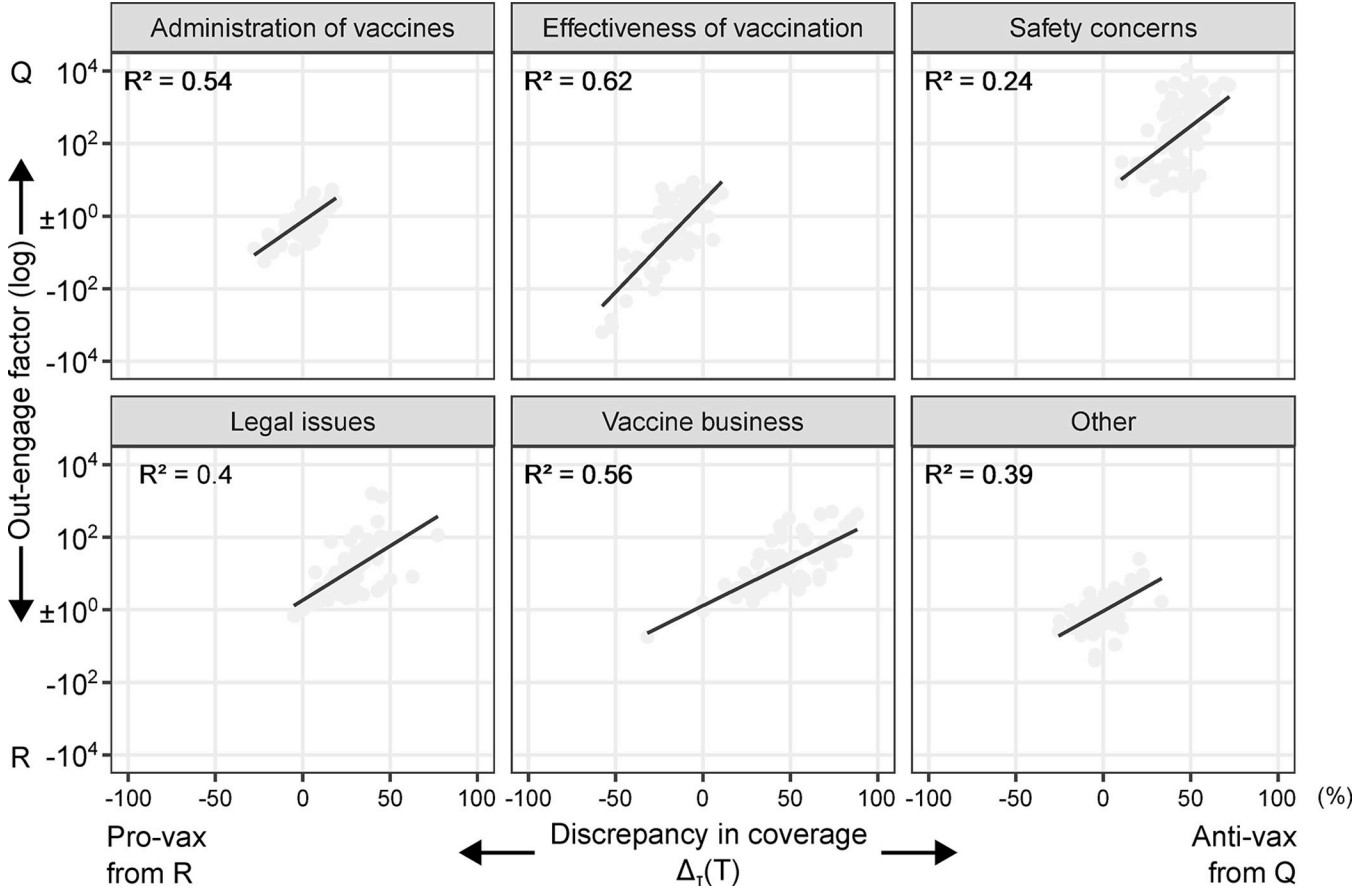

**Fig 4. Relationship between the discrepancy in coverage between anti-vax content from questionable sources and pro-vax content from reliable sources, and the corresponding out-engage factor for each topic.** The independent variable $\Delta_\tau(T)$ is positive if, during the month $T$ (spanning from January 2016 to December 2021), the anti-vax coverage of topic $\tau$ within source set Q exceeds the pro-vax coverage of $\tau$ within source set R, and negative otherwise. The dependent variable is positive if anti-vax coverage within Q overperforms pro-vax coverage within R in terms of user engagement, and negative otherwise. Solid lines and $R^2$ coefficients refer to log-linear regressions.

## Materials and methods

### Data collection

We first merged the lists from independent fact-checking organizations (i.e. bufale.net, butac. it, facta.news, newsguardtech.com, and pagellapolitica.it) to collect the main information providers in Italy among newspapers, online-only news outlets, radio stations, and TV channels. The news sources gathered have also been classified as questionable (whether the source has a reputation of regularly spreading misinformation) or reliable, according to the factualness classification they received. The final list of sources consists of:

- 96 out of the 121 major Italian newspapers that in 2021 reached 30 million Italians, i.e., $\sim 60\%$ of the population aged more than 18 (source: GfK Mediamonitor);

- 462 online-only news outlets that in 2021 monthly reached 40 million Italians, i.e., $\sim 96\%$ of the total internet audience (source: ComScore);

- 89 TV channels, including all RAI newscasts (3 national and 20 regional), that in 2021 monthly reached 8 million Italians, i.e., $\sim 54\%$ of the TV audience (source: Auditel);

- 35 radio stations that in 2021 daily reached 26 million Italians, i.e. $\sim$77% of radio listeners (source: RadioTER - Tavolo Editori Radio).

Except for one TV channel, which primarily operates online as well, all the remaining questionable sources (160) fall under the category of online-only news outlets.

This quasi-census approach applied to both questionable and reliable news sources enabled us to virtually capture the entirety of vaccine-related information provided to Italians in recent years (NewsGuard alone claims to monitor domains covering about 95% of online engagement with news sites [45]). Specifically, we collected all vaccine-related content published by these 682 sources on Facebook, Instagram, Twitter, and YouTube between 2016 and 2021, along with the corresponding user interactions (likes, comments, shares, etc.). It is worth noting that, with the only exception of the instant messaging services WhatsApp and Facebook Messenger, the four analyzed platforms represent the most used social media in Italy during 2021: YouTube was used by 85.3% of Internet users aged 16 to 64, Facebook 80.4%, Instagram 67%, and Twitter 32.8% (source: GWI). To do this, we searched for content whose textual parts (message, image text, or any other description) matched an exhaustive list of vaccine-related keywords, including general terms (e.g. vaccine, vaccination) and vaccine brands/names, both mandatory (e.g. Hexyon, Menjugate), recommended (e.g. Bnt162b2, Gardasil, Janssen, Twinrix) and others available (e.g. Vaxchora, Ervebo). The complete list of keywords was retrieved from the website of the Italian Medicines Agency (See S1 Table for details). Data from Facebook and Instagram were collected through CrowdTangle [46], a Facebook-owned tool that tracks interactions on public content from various social media platforms. Data from Twitter and YouTube were gathered by means of their official APIs. Twitter API was accessed through academic account before the limitations introduced by the new management.

Table 3 shows a breakdown of the vaccine dataset. Data are divided by source set and period analyzed (Pre-pandemic 01/01/2016–29/01/2020, Pandemic 30/01/2020–31/12/2021, Overall 01/01/2016–31/12/2021), and concern the number of sources, contents, and corresponding user interactions, understood as the algebraic sum of all possible actions/reactions performed on the four platforms analyzed (S1 Fig also shows the prevalence of misinformation on vaccines according to the focus of the news sources selected).

Aside from variables that uniquely identify a news item (e.g. content id, author id, date of creation, URL), or variables concerning its content (e.g. message, image text, content type), each observation in the vaccine dataset also includes the count of followers at posting. This information is crucial for calculating user engagement using Eq (3). See S6 Fig for the distribution of followers by category (Questionable vs. Reliable) across Facebook, Instagram, Twitter, and YouTube.

The various APIs utilized for collecting individual content also enable us to obtain time-series metrics for single sources or sets of sources. Hence, to get an accurate estimate of how much attention questionable sources and reliable sources have dedicated, respectively, to the topic of vaccines compared to the rest of the covered topics, we downloaded the time-series of the total contents published and the total interactions gained by the two source sets.

## Time-series and causality analysis

The correlation functions for testing and measuring causality (e.g. Granger causality [47]) have been applied in several fields, including social media [48,49]. Despite the widespread, their use is limited to linear relations, although linear models can not accurately represent real-world interactions. Further, while all they determine whether two time-series have correlated movement, no directional information about cause and effect can be inferred. On the contrary, information-theoretic approaches understand causality as a phenomenon that can

be not only detected or measured but also quantified. In addition, they are sensitive to nonlinear signal properties, as they do not rely on linear regression models.

In the analysis, we relied on the concept of Transfer Entropy (TE) to estimate the strength and direction of information transfer between the daily time-series of the percentage of vaccine-related content from Questionable and Reliable sources, respectively.

TE [38] is the model-free measure of a (Shannonian) information transfer defined by means of Kullback–Leibler divergence [50] on conditional transition probabilities $p$ of two Markov processes $X$ and $Y$ of orders $k$ and $l$, respectively, as

$$\text{TE}_{X \to Y}(k, l) = \sum_{x \in X, y \in Y} p\left(y_{t+1}, y_t^{(l)}, x_t^{(k)}\right) \log \frac{p\left(y_{t+1} | y_t^{(l)}, x_t^{(k)}\right)}{p\left(y_{t+1} | y_t^{(l)}\right)} \tag{1}$$

where $x_t^{(k)} = (x_t, \ldots, x_{t-k+1})$ and $y_t^{(k)} = (y_t, \ldots, y_{t-k+1})$. The estimate $TE_{Y \to X}(l,k)$ of the information transfer from $Y$ to $X$ is derived analogously. For independent processes, TE is equal to zero.

Since a straightforward implementation of Eq (1) could lead to biased estimates when the expected effect is rather small or the sample size is limited [51], we also calculated the Effective Transfer Entropy (ETE) [44] defined as

$$\text{ETE}_{X \to Y}(k, l) = \text{TE}_{X \to Y}(k, l) - \text{TE}_{X_{\text{shuffled}} \to Y}(k, l) \tag{2}$$

where $\text{TE}_{X_{\text{shuffled}} \to Y}(k, l)$ indicates the average transfer entropy over independently shuffled $X$.

To assess the statistical significance of Eq (2), we applied a bootstrap procedure of the Markov process underlying $X$ that destroys the statistical dependencies between $X$ and $Y$ but, conversely from only shuffling, retains the dependencies within $X$ [52]. ETE is calculated by using 100 shuffles and 300 bootstrap replications to obtain the distribution of the estimates under the null hypothesis of no information flow [53].

Common choices of the Markov block length $k$ in $\text{TE}_{X \to Y}(k,l)$ and $\text{TE}_{Y \to X}(k,l)$ are $k = l$ and $k = l$, and the last is usually preferred [38]. Thus, the analysis in the current study is conducted by setting $k = l = 1$ [41]. In other words, we measure the capacity of one time-series to predict the immediate future of the another, i.e. just one symbol ahead [48,54].

TE estimates are based on discrete data. Hence, we transformed our series into symbol sequences by partitioning the data into $m$ bins. Suitable values of $m$ have been empirically proven to be in the range [3,7,55]. Moreover, since in most cases $m > 5$ does not imply a better projection of the data in the symbol space, we consider $3 \leq m \leq 5$ [48]. In our case study, the highest daily percentage of vaccine-related content from both Questionable and Reliable source sets is $\sim 16\%$, hence we rely on powers of two for identifying the five bins (0,1], (1,2], (2,4], (4,8], (8,100] (See S4 Table for the bin-quantile correspondence).

## User engagement and overperforming content

Let $\mathcal{U}$ be a universe of new sources and $\emptyset \neq S \subset \mathcal{U}$. We denote by $C(S;T)$ and $I(S;T)$ the number of contents published by the whole $S$ in the time span $T$ and the corresponding number of user interactions, respectively. Let now $\mathcal{X}$ be a universe of pairwise disjoint features and $X \subset \mathcal{X}$. We write $C(S;X;T)$ and $I(S;X;T)$ for denoting that the quantities concern the set of features $X$.

We compute the total user engagement with the $X$-related content published by $S$ during $T$ as the real number:

$$E(S; X; T) = \frac{I(S; X; T)}{C(S; X; T) \cdot F(S; T)} \tag{3}$$

where $F(S;T)$ represents the average number of followers of the social media accounts of $S$ which were active during $T$. In other words, if $s \in S$ did not publish any content on any of the analyzed platforms throughout $T$, its contribution $E(s;X;T)$ to $E(S;X;T)$ is 0. If $s$ was only active on Facebook during $T$, then $F(s;T)$ counts only the average number of its fans on Facebook during $T$.

To assess the importance of the $X$-related content published by $S$ throughout $T$ in terms of user engagement, we investigate two different points of view: the out-engage factor of $X$ to $X^C$, that is the complement of $X$ in $\mathcal{X}$ (internal comparison); the out-engage factor of $X$ in $S$ to itself in $S^C$, that is the complement of $S$ in $\mathcal{U}$ (external comparison). Namely, we refer to the factor of proportionality of $E(S;X;T)$ to $E(S;X^C;T)$ in the former case, and to the factor of proportionality of $E(S;X;T)$ to $E(S^C;X;T)$ in the latter case. To these aims, we consider the function with codomain $\mathbb{R} \backslash ([-1,0] \cup (0,1])$ defined by

$$\mathcal{P}(S, S'; X, X'; T) = \delta(S, S'; X, X'; T) \left( \frac{E(S; X; T)}{E(S'; X'; T)} \right)^{\delta(S, S'; X, X'; T)} \tag{4}$$

where $S'$ is another set of sources, $X'$ another set of subjects, and $\delta$ stands for the sign function of the difference $E(S;X;T) - E(S';X';T)$:

$$\delta(S, S'; X, X'; T) = \begin{cases} 1 & \text{if } E(S; X; T) > E(S'; X'; T) \\ 0 & \text{if } E(S; X; T) = E(S'; X'; T) \\ -1 & \text{if } E(S; X; T) < E(S'; X'; T) \end{cases} \tag{5}$$

It is straightforward to notice that $\mathcal{P}(S, S'; X, X'; T) = 0$ if and only if the user engagement on $X$-related content from $S$ during $T$ equals the user engagement on $X'$-related content from $S'$ during the same time span. Otherwise, if $\mathcal{P}(S, S'; X, X'; T) \in (1, \infty]$ then the user engagement on $X$-related content from $S$ is higher than the user engagement on $X'$-related content from $S'$, and we say that $X$ is overperforming in $S$ with respect to $X'$ in $S'$ during $T$. Conversely, if $\mathcal{P}(S, S'; X, X'; T) \in [-\infty, -1)$ we say that $X'$ is overperforming in $S'$ with respect to $X$ in $S$ during $T$.

For $S' = S$ and $X' = X^C$ the value returned by Eq (4) responds to the internal comparison, and we simply write $\mathcal{P}(S; X, X^c; T)$. For $S' = S^C$ and $X' = X$ it responds to the external comparison, and we simply write $\mathcal{P}(S, S^c; X; T)$.

In our analysis, we partition the selected source set into questionable and reliable subsets, and then compare the distributions of the positive and negative daily out-engage factors related to the vaccine subject from both perspectives (internal and external). Limited to vaccine-related content, we also investigate both the perspectives in the universe of possible stances conveyed (anti-vax, neutral, pro-vax). The general Eq (4) is instead used for comparing the topic-specific engagement of anti-vax content from questionable sources and the pro-vax content from reliable sources.

Note that the news items collected were processed regardless of the social media where they were published. In other words, the contents $c(s;T)$ published by a news source s during the time span $T$ refer to the totality of its Facebook posts, Instagram media, Twitter tweets and YouTube videos. Analogously, the user interactions $i(s;T)$ is defined as sum of all actions taken

on $c(s;T)$ throughout $T$: comments, shares, likes and other reactions (angry, haha, love, sad, wow) on Facebook posts; comments and likes on Instagram media; replies, retweets and likes on Twitter tweets; comments, likes and dislikes on YouTube videos.

## Modelling stance conveyed and topic discussed in vaccine-related content

Despite the recent widespread adoption of Large Language Models (LLMs), when labeled data is available, fine-tuning a smaller LLM remains the preferred method for text classification [56]. Here, we choose Google BERT [57], which represents the state-of-the-art for semantic text representation in most languages [58], to fine-tune a model capable of predicting whether an Italian text conveys anti-vax, neutral, or pro-vax stance, as well as a model capable of predicting the specific topic discussed.

## Data selection, annotation, and augmentation

The content to be annotated were sampled from the collected data at a rate of approximately 10%. To get a training set as rich as possible with both anti-vax and pro-vax stance, we intentionally annotated about three-quarters (9,071) of content published by those news sources that mainly cover topics concerning medicine, science, and technology, both questionable (the more likely to convey anti-vax stance) and reliable (the more likely to convey pro-vax stance). Other 25,232 contents to be annotated were randomly selected from the data produced by the remaining sources. The data to annotate was split among the authors. The splitting procedure was optimized to get $\sim 20\%$ overlap between the authors. This allowed us to compare the annotator agreement results with the model performance (See Classification). The total annotated data consist of 34,303 contents, divided according to the stance conveyed in 9,902 anti-vax, 17,258 neutral, and 7,143 pro-vax.

Since anti-vax and pro-vax stances are only conveyed by about half of the annotated contents, we applied a text data augmentation technique to make the model more balanced between stance classes and more familiar with the local space around non-neutral positions. Namely, we relied on the nlpaug Python library [59] to get 11,712 augmented contents. Augmented contents were obtained by inserting words in a selection of data annotated as anti-vax or pro-vax through the contextual word embedding of BERT, i.e., the pre-trained language model then fine-tuned to the annotated data. The data to be augmented were chosen randomly but preserving the topic distribution of the whole annotated dataset.

The augmented dataset was then split into two parts to produce a dataset for training (80%) and a dataset for evaluating (20%) the model, by ensuring on both sets the same class distribution with respect to both stances and topics. To assure proper model evaluation, neither the annotated content used as a basis for the augmentation, nor the augmented content were included in the evaluation set.

The annotation process also concerned the identification of the topic discussed: one of administration of vaccines, vaccine business, effectiveness of vaccination, legal issues, safety concerns, other.

Table 4 summarizes the annotation results with respect to opinion and topic for the training and evaluation sets.

**Classification.** A state-of-the-art neural model based on Transformer language models was trained to distinguish between the three stance classes. We used the pre-trained BERT multilingual cased model [57] consisting of 12 stacked Transformer blocks with 12 attention heads each. We attached a linear layer with a softmax activation function at the output of these layers to serve as the classification layer. As input to the classifier, we take the representation of the special [CLS] token from the last layer of the language model. The whole model is jointly

**Table 4. Annotation results for both training (a) and evaluation (b) sets.** Rows refer to stance classes: A = anti-vax, N = neutral, P = pro-vax. Columns refer to topic classes: Adm = administration of vaccines, Bus = vaccine business, Eff = effectiveness of vaccination, Leg = legal issues, Saf = safety concerns, Oth = other.

**(a)** Training set.

|  | Adm | Bus | Eff | Leg | Oth | Saf | Σ | |
|---|---|---|---|---|---|---|---|---|
| A | 941 | 1,019 | 1,895 | 929 | 238 | 6,664 | 11,686 | (31%) |
| N | 6,733 | 311 | 1,816 | 1,379 | 1,121 | 2,351 | 13,711 | (38%) |
| P | 1,734 | 320 | 5,664 | 491 | 435 | 2,681 | 11,325 | (31%) |
| Σ | 9,408 | 1,650 | 9,375 | 2,799 | 1,794 | 11,696 | 36,722 | (100%) |
| | (26%) | (4%) | (25%) | (8%) | (5%) | (32%) | (100%) | |

**(b)** Evaluation set.

|  | Adm | Bus | Eff | Leg | Oth | Saf | Σ | |
|---|---|---|---|---|---|---|---|---|
| A | 235 | 254 | 474 | 232 | 59 | 1,666 | 2,920 | (31%) |
| N | 1,808 | 78 | 454 | 344 | 280 | 587 | 3,551 | (38%) |
| P | 433 | 80 | 1,415 | 122 | 108 | 670 | 2,828 | (31%) |
| Σ | 2,476 | 412 | 2,343 | 698 | 447 | 2,923 | 9,299 | (100%) |
| | (26%) | (4%) | (25%) | (8%) | (5%) | (32%) | (100%) | |

trained on the downstream task of three-class stance identification. According to the BERT reference paper, fine-tuning of the neural models was performed end-to-end. We used the Adam optimizer with the learning rate of $5e{-}5$ and weight decay set to 0.01 for regularization [60]. The model was trained for 4 epochs with batch size 64 through the HuggingFace Transformers library [61].

The same pre-trained architecture and hyperparameters were also used to train a model for distinguish between the six topics.

Table 5 reports the performance of the trained models compared with the inter-annotator agreement by using the same measure: accuracy (Acc) and the F1 score for individual classes, on both the training and the evaluation datasets. The confusion matrices for the evaluation set, used to compute all the scores of the annotator agreements and the model performance, are reported in S11 and S12 Tables.

The models are applied to all the collected data to classify them based on the conveyed stance and discussed topic, respectively.

## Conclusions

Communication plays a pivotal role in the representation of reality and thus in the formation of opinions and the orientation of individual behavior, especially on the web. The internet and social media platforms offer vast opportunities for user interaction but also serve as significant channels for the dissemination of inaccurate or intentionally deceptive information. This trend is especially detrimental when the subject of misinformation pertains to health, such as vaccines, as it can have profound repercussions on people's well-being and quality of life.

The proliferation of anti-vaccination misinformation on social media has heightened its urgency, particularly amidst the unprecedented scale of the Covid-19 pandemic and the urgent need for widespread vaccination efforts. Despite extensive research on the prevalence of health-related misinformation online, the full scope of this issue remains uncertain. Nevertheless, there is evidence suggesting that individuals' acceptance of online misinformation significantly influences their willingness to receive vaccines.

Through a comprehensive analysis of the social media news content produced by a nationally representative sample of TV, radio, print and online-only news outlets over a 6-year time

**Table 5. Performance of our stance (a) and topic (b) classification models on the training set and the evaluation set, in comparison to the inter-annotator agreement on the same datasets.** The overall performance is measured by accuracy (Acc), and performance for individual classes by F1 score.

**(a)** Stance model.

| Performance and agreement | Overall | A | N | P |
|---|---|---|---|---|
| | Acc | F1 | F1 | F1 |
| **Model** | | | | |
| Training | 0.93 | 0.93 | 0.91 | 0.94 |
| Evaluation | 0.88 | 0.88 | 0.86 | 0.90 |
| **Inter-annotator** | | | | |
| Training | 0.89 | 0.90 | 0.86 | 0.89 |
| Evaluation | 0.89 | 0.90 | 0.87 | 0.89 |

**(b)** Topic model.

| Performance and agreement | Overall | Adm | Bus | Eff | Leg | Saf | Oth |
|---|---|---|---|---|---|---|---|
| | Acc | F1 | F1 | F1 | F1 | F1 | F1 |
| **Model** | | | | | | | |
| Training | 0.94 | 0.94 | 0.85 | 0.94 | 0.88 | 0.95 | 0.79 |
| Evaluation | 0.88 | 0.88 | 0.83 | 0.89 | 0.80 | 0.92 | 0.73 |
| **Inter-annotator** | | | | | | | |
| Training | 0.91 | 0.89 | 0.83 | 0.92 | 0.81 | 0.95 | 0.78 |
| Evaluation | 0.87 | 0.86 | 0.78 | 0.89 | 0.78 | 0.92 | 0.70 |

span, we shed light on the real impact of vaccine misinformation on both the information available to social media users and their news diet.

Our results highlight a complex picture that needs to be illustrated in all its facets. Although we find misinformation making up a relatively small but not insignificant (12.6%) part of all the news content supplied during the period 2016–2021, information dynamics shifts over time, with the percentage of misinformation nearly tripling (31.7%) when focusing on the period before the Covid-19 outbreak. This increased prevalence of misinformation also coincides with a more significant information flow from questionable to reliable sources than in the opposite direction, framing misinformation as driver of the public debate on vaccines. Striking results also arise from comparing user engagement with vaccine-related content produced by misinformation and non-misinformation sources, respectively, for which a normalization by followers is very necessary to control for possible scaling effects. Our analysis returns a median engagement 6 times higher for misinformation than non-misinformation during the overall period, which rises to 11 when time is limited to before Covid-19 outbreak.

While these results show the prominent role achieved by misinformation sources in the news ecosystem, the pandemic shock confirms the detrimental effects of the convulsive dynamics of the public agenda on social debates. The issue-attention cycle [62] and the consequent need to continuously emphasize trending topics (the pre-pandemic period includes the 2016 US presidential election, the 2016 Italian constitutional referendum, the succession of two legislatures (XVII and XVII) and four governments (Renzi, Gentiloni, Conte I, Conte II), and important news events, such as the murder of Giulio Regeni, the 2016–2017 Central Italy earthquakes, the Morandi Bridge collapse, and many others) shorten the amount of time available to discuss each matter—especially those that may have a negative impact on societies— and prevent online audiences from engaging in a thoughtful public debate [63]. The Covid-19 pandemic has been an unprecedented event, not just from an epidemiological perspective, but also for the entire information ecosystem. Since the onset of 2020 and spanning over two years, news regarding the virus, including discussions about potential vaccines, has profoundly

impacted almost every facet of media production, unlike any other event in recent history. Consequently, misinformation sources have lost their leading role in the public debate on vaccines and have seen a substantial reduction in the engaging power they once exhibited prior to the Covid-19 outbreak.

Despite the exceptional nature of the Covid-19 event, the spread ease of false claims is only partially attributable to the presence of misinformation sources, and more likely due to the inability of mainstream media to drive the public debate over time on issues that are particularly sensitive and emotional. In other words, to properly account for the temporal dynamics of public debate is crucial to prevent the latter from moving into uncontrolled spaces where unreliable information is more easily conveyed, potentially exacerbating vaccine hesitancy among the users involved. By leveraging on state-of-the-art deep learning models capable of accurately classifying vaccine-related content based on conveyed stance and discussed topic, respectively, we demonstrate that this trend mainly concerns anti-vax rhetoric on the most sensitive topics, namely, vaccine effectiveness and safety. At the same time, our results confirm the efficacy of assiduously proposing a convincing counter-narrative to misinformation spread [64]. Namely, the effectiveness of vaccination, which reliable sources have adequately promoted, appears to be the topic least affected by anti-vax rhetoric in terms of user engagement. Conversely, insufficient coverage of vaccine safety by pro-vax sources correlates with the highest engagement with misinformation content conveying an anti-vax stance.

## Supporting information

**S1 Fig. Prevalence of misinformation by source scope.** By exploiting the category classification provided by CrowdTangle and inspecting the page descriptions on Facebook, we extract the primary focus of all the 682 sources selected. Similar categories or those belonging to the same thematic area have been grouped together to streamline and condense the category list (e.g. pages dedicated to the topics of soccer and basketball, respectively, have merged into the Sports category). The final range of categories includes General News, Business/Finance, Culture & Society, Entertainment, Lifestyle, Nature & Animals, Politics, Religion, Science & Technology, Sports, Weather. Table shows the prevalence of misinformation on vaccines in relation to these categories. The color of the bars is associated with the total number of contents present in our dataset, regardless of the reliability of the source that produced them. Although General News is the primary focus for the questionable sources that produced most of the content on vaccines, these sources only contribute to one-tenth of the total vaccine-related content compared to reliable sources. Among specialized topics, while Entertainment, Sports, and Religion are rarely the main focus of vaccine misinformation, pages dedicated to Lifestyle, Nature & Animals, and Weather exclusively disseminate misinformation about vaccines. A clear prevalence of questionable sources is also evident in the Politics and Culture & Society categories, while a more balanced distribution is observed in the Business/Finance and Science & Technology categories.
(TIF)

**S2 Fig. Prevalence of vaccine misinformation on Facebook, Instagram, Twitter, and YouTube.** Plot shows the daily time-series, depicting the proportion of vaccine-related content originating from questionable sources in relation to the total volume of vaccine-related content (both from questionable and reliable sources). To bring out trends more clearly, for each social media, the time-series displayed concerns a 30-days simple moving average. Facebook stands out as the social media platform where vaccine misinformation is most prevalent. In the pre-pandemic period, approximately one out of every two vaccine-related contents published on the platform originates from questionable sources. In the subsequent period, the increased

attention on the topic from reliable sources has brought the proportion back to more sustainable levels. Inset shows the quantity of vaccine-related content generated by the selected sources on each of the four social media platforms. Instagram and YouTube exhibit values that are one order of magnitude lower than those on Facebook and Twitter.
(TIF)

**S3 Fig. Daily time-series of the percentage of vaccine-related content from questionable and reliable sources, respectively, and associated first difference.** Graphics are broken down by period: Overall (1 January 2016–31 December 2021), pre-pandemic (1 January 2016–29 January 2020) and pandemic (30 January 2020–31 December 2021).
(TIF)

**S4 Fig.** Cross-correlation function (CCF), i.e., ratio of covariance to root-mean variance, for daily time-series of the percentage of vaccine-related content from questionable (left panel) and reliable (right panel) sources, respectively. Plots concern the overall period (1 January 2016–31 December 2021) and the pre-pandemic (1 January 2016–29 January 2020) and pandemic (30 January 2020–31 December 2021) sub-periods, respectively. In the left (right) panel the lagged values refer to questionable (reliable) time-series.
(TIF)

**S5 Fig. Percentage of vaccine coverage with respect to the different stance classes (A = Anti-vax, N = Neutral, P = Pro-vax).** The observations underlying each empirical Probability Density Function curve represent the single sources and their sizes the corresponding amount of vaccine coverage.
(TIF)

**S6 Fig. Boxplot of follower distribution by category (Questionable vs. Reliable) across different platforms (Facebook, Instagram, Twitter, YouTube).** Outliers have been removed, and the y-axis range is limited to the 5th and 95th percentiles to highlight the central tendency of follower distribution within each platform. Median follower counts: Facebook 31,550 (Q), 50,088 (R); Instagram 4,736 (Q), 14,080 (R); Twitter 1,461 (Q), 6,883 (R); YouTube 11,736 (Q), 2,903 (R). The number of observations (i.e., sources) is indicated at the base of each boxplot.
(TIF)

**S1 Table. List of keywords used for filtering vaccine-related content.** Table reports keywords divided by vaccine type: General, Covid-19 vaccines, Mandatory and Recommended vaccines (Law 119/2017), Other authorized and marketed vaccines. The special character * stands for zero or more non-whitespace characters. The list does not include the term 'immunizzazione' (immunization) and its variations, since anti-vaccination contents almost never talks about immunization [65,66]. Obviously, this is expected since anti-vaccination groups tend not to believe that vaccines confer immunity. Possible uses of the searched keywords in contexts other than the discussion on vaccines are, of course, quite rare. However, in the manual annotation process concerning stance and topic, we observed that the only critical cases involved content containing references to 'latte vaccino' (vaccine milk). Such content has been removed from the dataset.
(DOCX)

**S2 Table. Descriptive statistics for the daily time-series (TS) of the percentage of vaccine-related content from questionable and reliable sources, respectively, and the corresponding first difference (FD).** The measures are reported for the overall sample (1 January 2016–31 December 2021), together with the pre-pandemic (1 January 2016–29 January 2020) and

pandemic (30 January 2020–31 December 2021) sub-periods.
(DOCX)

**S3 Table. Augmented Dickey Fuller (ADF) test statistics for daily time-series of the per-centage of vaccine-related content from questionable and reliable sources, respectively.**
The table reports the t-statistic of the ADF test statistics both in levels and on the first difference (FD). The ADF test is based on regressions with intercept. The null hypothesis for the test is non-stationarity. The estimates are reported for the overall period (1 January 2016–31 December 2021), together with the pre-pandemic (1 January 2016–29 January 2020) and pandemic (30 January 2020–31 December 2021) sub-periods.
(DOCX)

**S4 Table. Correspondence bin upper bound—quantile for questionable and reliable time-series, respectively.** Bins refer to the symbolic encoding performed to calculate the transfer entropy. The measures reported refer to both the overall sample (1 January 2016–31 December 2021), and the pre-pandemic (1 January 2016–29 January 2020) and pandemic (30 January 2020–31 December 2021) sub-periods.
(DOCX)

**S5 Table. Comparison of engagement gained by vaccine-related content within question-able sources and within reliable sources, respectively (internal comparison).** Denoted with $X$ the vaccine subject and with $X^C$ the totality of other subjects covered during day $d$, Table shows the results of Mann-Whitney U test applied to the distributions of the out-engage factor $P(Q;X,X^C;d)(P(R;X,X^C;d))$ for the days $d$ when it is in favor of the vaccine subject compared to the rest of subjects discussed within the source set Q (R) and vice versa, respectively. Since the distributions have values of opposite sign (See Eq (4) in the main text), the test is applied to the distributions of absolute values. Distributions are compared according to the period analyzed: Overall (1 January 2016–31 December 2021), pre-pandemic (1 January 2016–29 January 2020) and pandemic (30 January 2020–31 December 2021).
(DOCX)

**S6 Table. Comparison of engagement gained by vaccine-related content from source set Q to source set R (external comparison).** Denoted with $X$ the vaccine subject, Table shows the results of Mann-Whitney U test applied to the distributions of the out-engage factor $P(Q,R;X;d)$ for the days $d$ when it is in favor of source set Q and source set R, respectively. Since the distributions have values of opposite sign (See Eq (4) in the main text), the test is applied to the distributions of absolute values. Distributions are compared according to the period analyzed: Overall (1 January 2016–31 December 2021), pre-pandemic (1 January 2016–29 January 2020) and pandemic (30 January 2020–31 December 2021).
(DOCX)

**S7 Table. Comparison of engagement gained by vaccine-related content expressing the corresponding stance to vaccine-related content conveying any other stance, within ques-tionable sources and within reliable sources, respectively (internal comparison).** Denoted with $X$ the vaccine subject covered through one of the three stances by source set Q (R) and with $X^C$ the vaccine subject covered through the other two stances by the same source set, Table shows the results of Mann-Whitney U test applied to the distributions of the out-engage factor $P(Q;X,X^C;d)(P(R;X,X^C;d))$ for the days $d$ when it is in favor of the vaccine subject compared to the rest of subjects discussed within the source set Q (R) and vice versa, respectively. Since the distributions have values of opposite sign (See Eq (4) in the main text), the test is applied to the distributions of absolute values. Distributions are compared according to the

period analyzed: Overall (1 January 2016–31 December 2021), pre-pandemic (1 January 2016–29 January 2020) and pandemic (30 January 2020–31 December 2021).
(DOCX)

**S8 Table. Comparison of engagement gained by vaccine-related content expressing the corresponding stance from source set Q to source set R (external comparison).** Denoted with $X$ the vaccine subject covered through one of the three stances, Table shows the results of Mann-Whitney U test applied to the distributions of the out-engage factor $P(Q,R;X;d)$ for the days $d$ when it is in favor of source set Q and source set R, respectively. Since the distributions have values of opposite sign (See Eq (4) in the main text), the test is applied to the distributions of absolute values. Distributions are compared according to the diverse stance conveyed (anti-vax, neutral, or pro-vax) and the period analyzed: Overall (1 January 2016–31 December 2021), pre-pandemic (1 January 2016–29 January 2020) and pandemic (30 January 2020–31 December 2021).
(DOCX)

**S9 Table. Parameters of the log-linear model describing the relationship between the discrepancy in coverage between anti-vax content from questionable sources and pro-vax content from reliable sources, and the corresponding out-engage factor for each topic (Adm = administration of vaccines, Bus = vaccine business, Eff = effectiveness of vaccination, Leg = legal issues, Saf = safety concerns, Oth = other).** Reported data are estimates for intercept ($\alpha$) and slope ($\beta$), corresponding confidence interval (95%) and significance, number of observations (months in the analyzed period), $R^2/R^2adj$.
(DOCX)

**S10 Table. Percentage distribution of vaccine-related content among the six topics identified (Adm = administration of vaccines, Bus = vaccine business, Eff = effectiveness of vaccination, Leg = legal issues, Saf = safety concerns, Oth = other).** Data are divided by source category (questionable and reliable) and period analyzed (Overall 1 January 2016–31 December 2021, Pre-pandemic 1 January 2016–29 January 2020, Pandemic 30 January 2020–31 December 2021). Percentages are further divided according to the stance conveyed by the corresponding content (A = anti-vax, N = neutral, P = pro-vax).
(DOCX)

**S11 Table.** Confusion matrices for the evaluation set with respect to stance: Between the annotators and the model (a), and between annotators (b). The performance measures, Acc and F1, are calculated from these matrices. The axes show the possible labels (A = anti-vax, N = neutral, P = pro-vax).
(DOCX)

**S12 Table.** Confusion matrices for the evaluation set with respect to topic: Between the annotators and the model (a), and between annotators (b). The performance measures, Acc and F1, are calculated from these matrices. The axes show the possible labels (Adm = administration of vaccines, Bus = vaccine business, Eff = effectiveness of vaccination, Leg = legal issues, Saf = safety concerns, Oth = other).
(DOCX)

**S13 Table. Modelling the interplay between news reporting and user engagement through random effects regressions with AR(1) disturbance.** Results refer to the model
$y_{it} = \alpha + \beta x_i + \gamma z_{it} + \theta_i + \epsilon_{it}, \quad i = 1, 2, \ldots, N, \ t = 1, 2, \ldots, T$, where $i$ is the news source identifier, $t$ is time (days since January 1st, 2016), $y_{it}$ is the user engagement on vaccine-related content from source $i$ on day $t$, $x_i$ is a vector of time-independent source-related variables (e.g.,

the factualness classification: Reliable or questionable), $z_{it}$ are time-dependent factors (e.g., number, type and topic of news content), and $\theta_i, \epsilon_{it}$ are error terms with auto-correlated component. Being the panel data unbalanced and $T>N$, we choose a model with autocorrelated disturbances. Robust estimators of variance were also estimated, and the results are confirmed [67,68]. Model I results show that content conveying pro-vax stance decreases user engagement (elasticity -0.03%) while anti-vax stance increases it (elasticity 0.5%), especially when it is proposed by questionable sources (Model II, coefficient of the interactive variable 1.34%). Conversely, when questionable sources convey pro-vax stance, engagement drops by 0.3%. If we focus on topics (Models III and IV), "safety concerns" is the one that elicits the greatest reactions (0.4%), especially when content comes from questionable sources (maximum elasticity 1.1%). Engagement drops instead when questionable sources cover the topic of effectiveness of vaccination.
(DOCX)

## Acknowledgments

The authors thank Luciano Pietronero, Vittorio Loreto, Serge Galam, Alessandro Galeazzi, Antonio Scala, and Fabiana Zollo for suggestions and comments on earlier versions of the article.

## Author Contributions

**Conceptualization:** Emanuele Brugnoli, Marco Delmastro.

**Data curation:** Emanuele Brugnoli.

**Formal analysis:** Emanuele Brugnoli, Marco Delmastro.

**Investigation:** Emanuele Brugnoli, Marco Delmastro.

**Methodology:** Emanuele Brugnoli, Marco Delmastro.

**Software:** Emanuele Brugnoli, Marco Delmastro.

**Supervision:** Emanuele Brugnoli, Marco Delmastro.

**Validation:** Emanuele Brugnoli, Marco Delmastro.

**Visualization:** Emanuele Brugnoli.

**Writing – original draft:** Emanuele Brugnoli, Marco Delmastro.

**Writing – review & editing:** Emanuele Brugnoli.

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
