## [Decision Letter · Decision Letter 0]

15 Mar 2023

PONE-D-22-33837

Dynamics of (mis)information flow and engaging power of narratives

PLOS ONE

Dear Dr. Brugnoli,

Thank you for submitting your manuscript to PLOS ONE. After careful consideration, we feel that it has merit but does not fully meet PLOS ONE’s publication criteria as it currently stands. Therefore, we invite you to submit a revised version of the manuscript that addresses the points raised during the review process.

ACADEMIC EDITOR:

Please address the comments provided by the reviewers.

We look forward to receiving your revised manuscript.

Kind regards,

Ankit Gupta

Academic Editor

PLOS ONE

Journal Requirements:

2. In your Methods section, please include additional information about your dataset and ensure that you have included a statement specifying whether the collection and analysis method complied with the terms and conditions for the source of the data.

Reviewers' comments:

Reviewer's Responses to Questions

**Comments to the Author**

1. Is the manuscript technically sound, and do the data support the conclusions?

Reviewer #1: No

Reviewer #2: Partly

Reviewer #3: Yes

2. Has the statistical analysis been performed appropriately and rigorously? 

Reviewer #1: Yes

Reviewer #2: Yes

Reviewer #3: I Don't Know

3. Have the authors made all data underlying the findings in their manuscript fully available?

Reviewer #1: Yes

Reviewer #2: Yes

Reviewer #3: Yes

4. Is the manuscript presented in an intelligible fashion and written in standard English?

Reviewer #1: No

Reviewer #2: Yes

Reviewer #3: Yes

5. Review Comments to the Author

Reviewer #1: In short, the authors need to dig in a lot more to the methods to better understand the biases introduced in their data collection on both production and consumption. Conditional on that, I find the key findings in Figure 3 and 4 compelling, but really need the authors to update their figure before I can really confirm. I understand the style of moving methods into a follow-up section after the discussion, but the paper is unreadable due to too much key parts in the methods section.

1) I am still very confused about the core data collection methods, even after re-reading the introduction and methods several times. What I think the authors did was go onto Crowtangle and the API for Twitter and YouTube and search for keywords around vaccines. They then took content that was from publications (discarding any posts) and graded quality or not based on publisher level. But, Figure 1 shows percent of all content? So, maybe they collect all links? Either way, way: what are they doing to validate their keywords (recall and precision of their labelling). Presumably they get some idea of false positives when hand coding for topic and position. And, what are they doing to deal with issues of images and videos.

2) The authors are using Facebook, Instagram, Twitter, and YouTube to measure engagement with content: what exactly are they measuring? In the methods they recognize that these platforms are measuring very different types of engagement, and they are measuring them differently. For example, CrowdTangle only includes URL with greater than 100 public shares. What the author are capturing is very different from views and biased, in many ways, towards click-bait or more sensational, low quality information. The authors should address this!

3) In Figure 1 (and assorted tables): percent of all content is highly related to how they are defining the corpus. This is especially important for reliable content: are the authors including sports and entertainment sections of the papers? Questionable content is more likely constrained to politically oriented publications?

4) For Figure 2 the authors are trying to measure how engaging vaccine related content is within a publication compared to other content, and within vaccine related content by follower. I find the terms confusing, but that is obviously not critical. The discovery of content is conditional on engagement making these measures troubled. For an article to be discovered through CrowdTangle it had to be publicly shared 100+ times! For the “outside” perspective, the authors are using “followers”, presumably on the platform? How well does this corelate with consumption and engagement? I assume low quality publications get followed very differently than high-quality publications.

5) The model is predicting the labeled position (1 of 3) with 88% in the evaluation set? That is sort of shockingly good. Conditional on all my previous concerns over the corpus, I find the left side of Figure 3 engaging and meaningful.

6) Similarly, accepting a sufficient level of accuracy, Figure 4 is also has the potential be meaningful. Interesting to see the different topics by publication type.

Reviewer #2: By identifying impact of misinformation authors are contributing a good quality of research but few things needs to be considered while representation of the paper.

1. Highlighting the state of the art contribution is important.

2. Abstract is not summarizing the exact unique contribution and goal of the work is not clear from abstract as well as introduction.

3. It requires strengthening paper presentation in several parts, including an introduction (Where is the novelty of the methods, explain it clearly)

4. The proposed model should include the proposed formulation

5. more emphasis is required on how the use of Deep learning model contributes to achieve results.

6. The authors should add the conclusion part with all the descriptions and results of this article.

7. The author can improve this paper by adding more current and fit references.

Reviewer #3: The paper seems very sound in its scientific method. It is also clearly written apart from a few places where the English could be improved (eg p7 line150). My concern has to do with the conceptual framework. In its title the paper claims to engage with "the power of narratives", yet while there are references to scientific literature dealing with "false narratives", nowhere does it offer anything like a systematic theory of narrative. It is hard to know where to begin in offering feedback on how to address this shortcoming, as there are so many different theories of narrative. The authors might consult Tangherlini et al https://doi.org/10.1371/journal.pone.0233879 which offers a good example of how a scientific publication for this journal can manage to combine a full discussion of its quantitative experimental method, while at the same time scientifically laying out a (qualitative) theory narrative, and showing how it applies that theory to its quantitative experiment. This, I think, is the standard to which the journal aspires. Finally, a smaller issue has to do with the opening conceptual framework of 'the public sphere', which I do not see as informing the analysis. The primary reference is 60 years old, with the more secondary reference being to a somewhat obscure Marxist figure. There has been so much written on this framework, that I question the need for opening this can of worms at all, when it seems like the paper is more interested in (polarized) political opinion than in the normative public sphere. As I say, the core problem is one of clarifying the framework, with the biggest issue pertaining to having an actual the theory of narrative, and secondarily getting clearer on the whole business of the public sphere and how that highly normative argument informs the argument of the paper as a whole—if the authors want to hold onto the latter, at the very least they'd need to come back to it in the conclusion.

6. PLOS authors have the option to publish the peer review history of their article (what does this mean?). If published, this will include your full peer review and any attached files.

Reviewer #1: No

Reviewer #2: No

Reviewer #3: No

---

## [Author Response · Author response to Decision Letter 0]

22 Apr 2024

Responses to specific reviewer and editor comments are already included in the attached files ("cover_letter.docx" for the Editor and "Response to Reviewers.docx" for the Reviewers).

---

## [Decision Letter · Decision Letter 1]

11 Sep 2024

PONE-D-22-33837R1Dynamics and triggers of misinformation on vaccinesPLOS ONE

Dear Dr. Brugnoli,

Thank you for submitting your manuscript to PLOS ONE. After careful consideration, we feel that it has merit but does not fully meet PLOS ONE’s publication criteria as it currently stands. Therefore, we invite you to submit a revised version of the manuscript that addresses the points raised during the review process.

The article is considerably implemented compared to its first version. However, a further revision step is necessary in order to further improve the methodological and argumentative aspects (see comments below). 

We look forward to receiving your revised manuscript.

Kind regards,

Andrea Cioffi

Academic Editor

PLOS ONE

Reviewers' comments:

Reviewer's Responses to Questions

**Comments to the Author**

1. If the authors have adequately addressed your comments raised in a previous round of review and you feel that this manuscript is now acceptable for publication, you may indicate that here to bypass the “Comments to the Author” section, enter your conflict of interest statement in the “Confidential to Editor” section, and submit your "Accept" recommendation.

Reviewer #1: (No Response)

Reviewer #4: (No Response)

2. Is the manuscript technically sound, and do the data support the conclusions?

Reviewer #1: Partly

Reviewer #4: Yes

3. Has the statistical analysis been performed appropriately and rigorously? 

Reviewer #1: Yes

Reviewer #4: I Don't Know

4. Have the authors made all data underlying the findings in their manuscript fully available?

Reviewer #1: Yes

Reviewer #4: Yes

5. Is the manuscript presented in an intelligible fashion and written in standard English?

Reviewer #1: Yes

Reviewer #4: Yes

6. Review Comments to the Author

Reviewer #1: 1) Why do the authors use 96 of 121 major Italian newspapers, rather than all 121? Can the authors break out this bulleted breakdown in Questionable and Reliable as shown in Table 1? I am most interested to know if all questionable come from online news or not?

2) I am not sure how I am supposed to think of transfer entropy in the pre-pandemic period. I do not know much about Italian vaccine laws, but judging by Figure 1, questionable sources start shooting up on coverage at a very similar time to the reliable sources, peak a still very low number, then peak after the law. For the rest of the time the value is near 0, to the point where inferences are basically built on noise? Meanwhile questionable sources have a valley between their peaks and then drop down before the reliable’ s second peak. I just want to understand if this time period, were all the action is, is driving the entropy results, or are they picking up on the relatively tiny movements over the rest of the timeframe? Either way, it may may sense to isolate that time period more, rather than the full pre-COVID period. I say this assuming that in the pre-COVID time period most of the people consuming anti-vax material were seeking it out with likely high personal demand for the material, making the production less interesting than the in the two times of acute demand during the new law and COVID.

3) E = interactions / (content * followers), and the authors use variations of this to see how content about vaccines and non-vaccines categorization, and/or between publications (or source types) within category do.

3a) I had to spend way too long to figure out exactly what the authors were doing, and how to interpret the plots, and I am still not sure I am reading everything correctly. In the writing the authors are leaning into differential interactions per content & followers, but they have created a pretty convoluted way to do it. I would like the authors to tie the findings in the paper directly to what they are referring to in the figures. I spent too much time searching around, especially as they are referring to specific values that are not directly noted on the charts.

3b) I assume many of the questionable content providers have very low follower counts, which could really impact the key findings?

3c) I assume higher interactions per content/followers were happening pre-pandemic because the content was more uniquely sensational, then the what was being produced during the pandemic (i.e., it is influenced both the much lower quantity of content and the click-baity nature of it). Could definitely think more about the crowding-out issues that happen during the pandemic.

I really like the work. The modelling is compelling. And the findings are pretty straight forward descriptive stats, hidden by a needlessly complex framework, that kept making me go back and forth between the methods, writing, and plots.

Reviewer #4: I find this manuscript very interesting and usefull for the scientific community involved in the study and promotion of the adequate use of vaccines. I am not an exprt in mathematical or deep learning models used in this paper, so I cannot evaluate the methodology used. To my best knowledge, the problem had a correct approach and the results provide important information regarding ways to get involved in proposing convincing counter-narratives to misinformation spread. Very important the results regarding the lack of engagement in the rhetoric of vaccine safety. We all should participate and learn from this study.

7. PLOS authors have the option to publish the peer review history of their article (what does this mean?). If published, this will include your full peer review and any attached files.

Reviewer #1: No

Reviewer #4: No

---

## [Author Response · Author response to Decision Letter 1]

24 Oct 2024

Responses to editor and reviewers are all included in the file labeled "Response to Reviewers".

---

## [Decision Letter · Decision Letter 2]

9 Dec 2024

Dynamics and triggers of misinformation on vaccines

PONE-D-22-33837R2

Dear Dr. Brugnoli,

We’re pleased to inform you that your manuscript has been judged scientifically suitable for publication and will be formally accepted for publication once it meets all outstanding technical requirements.

Kind regards,

Andrea Cioffi

Academic Editor

PLOS ONE

Additional Editor Comments (optional):

No further revisions are necessary.

Reviewers' comments:

Reviewer's Responses to Questions

**Comments to the Author**

1. If the authors have adequately addressed your comments raised in a previous round of review and you feel that this manuscript is now acceptable for publication, you may indicate that here to bypass the “Comments to the Author” section, enter your conflict of interest statement in the “Confidential to Editor” section, and submit your "Accept" recommendation.

Reviewer #1: All comments have been addressed

2. Is the manuscript technically sound, and do the data support the conclusions?

Reviewer #1: Yes

3. Has the statistical analysis been performed appropriately and rigorously? 

Reviewer #1: Yes

4. Have the authors made all data underlying the findings in their manuscript fully available?

Reviewer #1: Yes

5. Is the manuscript presented in an intelligible fashion and written in standard English?

Reviewer #1: Yes

6. Review Comments to the Author

Reviewer #1: (1) Old figure 2 is still in the manuscript, although I believe it is no longer in the paper.

(2) I still think the new Figure 2 is confusing, but I will leave that up to the editor if they think it is adequate for publication.

7. PLOS authors have the option to publish the peer review history of their article (what does this mean?). If published, this will include your full peer review and any attached files.

Reviewer #1: No

---

## [Editor Report · Acceptance letter]

17 Dec 2024

PONE-D-22-33837R2 

PLOS ONE

Dear Dr. Brugnoli, 

I'm pleased to inform you that your manuscript has been deemed suitable for publication in PLOS ONE. Congratulations! Your manuscript is now being handed over to our production team.

Kind regards, 

on behalf of

Dr. Andrea Cioffi 

Academic Editor

PLOS ONE